# Validation of a New Prognostic Score in Patients with Ovarian Adenocarcinoma

**DOI:** 10.3390/medicina59020229

**Published:** 2023-01-26

**Authors:** Oana Gabriela Trifanescu, Radu Iulian Mitrica, Laurentia Nicoleta Gales, Serban Andrei Marinescu, Natalia Motas, Raluca Alexandra Trifanescu, Laura Rebegea, Mirela Gherghe, Dragos Eugen Georgescu, Georgia Luiza Serbanescu, Haj Hamoud Bashar, Serban Dragosloveanu, Daniel Alin Cristian, Rodica Maricela Anghel

**Affiliations:** 1Discipline of Oncology, “Carol Davila” University of Medicine and Pharmacy, 022328 Bucharest, Romania; 2Department of Radiotherapy, “Prof. Dr. Al. Trestioreanu” Institute of Oncology, 022328 Bucharest, Romania; 3Department of Oncology, “Prof. Dr. Al. Trestioreanu” Institute of Oncology, 022328 Bucharest, Romania; 4Department of Surgery, “Prof. Dr. Al. Trestioreanu” Institute of Oncology, 022328 Bucharest, Romania; 5Department of Thoracic Surgery, “Prof. Dr. Al. Trestioreanu” Institute of Oncology, 022328 Bucharest, Romania; 6Discipline of Thoracic Surgery, “Carol Davila” University of Medicine and Pharmacy, 022328 Bucharest, Romania; 7Discipline of Endocrinology, “Carol Davila” University of Medicine and Pharmacy, 011863 Bucharest, Romania; 8“C.I. Parhon” Institute of Endocrinology, 011863 Bucharest, Romania; 9Discipline of Oncology, Faculty of Medicine and Pharmacy, “Dunarea de Jos” University, 800010 Galati, Romania; 10Department of Nuclear Medicine, “Prof. Dr. Al. Trestioreanu” Institute of Oncology, 022328 Bucharest, Romania; 11“Dr. Ion Cantacuzino” Department of Surgery, “Carol Davila” University of Medicine and Pharmacy, 030167 Bucharest, Romania; 12Department for Gynecology, Obstetrics and Reproductive Medicine, Saarland University Hospital, 66421 Homburg, Germany; 13Discipline of Orthopedics, “Foisor” Orthopedics Hospital, “Carol Davila” University of Medicine and Pharmacy, 022328 Bucharest, Romania; 14Discipline of Surgery, Coltea Clinical Hospital, “Carol Davila” University of Medicine and Pharmacy, 022328 Bucharest, Romania

**Keywords:** prognostic factors, prognostic score, ovarian carcinoma, adenocarcinoma

## Abstract

*Background and Objectives:* This study aimed to assess the impact of clinical prognostic factors and propose a prognostic score that aids the clinician’s decision in estimating the risk for patients in clinical practice. *Materials and Methods:* The study included 195 patients diagnosed with ovarian adenocarcinoma. The therapeutic strategy involved multidisciplinary decisions: surgery followed by adjuvant chemotherapy (80%), neoadjuvant chemotherapy followed by surgery (16.4%), and only chemotherapy in selected cases (3.6%). *Results:* After a median follow-up of 68 months, in terms of progression-free survival (PFS) and overall survival (OS), Eastern Cooperative Oncology Group (ECOG) performance status of 1 and 2 vs. 0 (hazard ratio—HR = 2.71, 95% confidence interval—CI, 1.96–3.73, *p* < 0.001 for PFS and HR = 3.19, 95%CI, 2.20–4.64, *p* < 0.001 for OS), menopausal vs. premenopausal status (HR = 2.02, 95%CI, 1.35–3,0 *p* < 0.001 and HR = 2.25, 95%CI = 1.41–3.59, *p* < 0.001), ascites (HR = 1.95, 95%CI 1.35–2.80, *p* = 0.03, HR = 2.31, 95%CI = 1.52–3.5, *p* < 0.007), residual disease (HR = 5.12, 95%CI 3.43–7.65, *p* < 0.0001 and HR = 4.07, 95%CI = 2.59–6.39, *p* < 0.0001), and thrombocytosis (HR = 2.48 95%CI = 1.72–3.58, *p* < 0.0001, HR = 3.33, 95%CI = 2.16–5.13, *p* < 0.0001) were associated with a poor prognosis. An original prognostic score including these characteristics was validated using receiver operating characteristic (ROC) curves (area under the curve—AUC = 0.799 for PFS and AUC = 0.726 for OS, *p* < 0.001). The median PFS for patients with none, one, two, three, or four (or more) prognostic factors was not reached, 70, 36, 20, and 12 months, respectively. The corresponding median overall survival (OS) was not reached, 108, 77, 60, and 34 months, respectively. *Conclusions:* Several negative prognostic factors were identified: ECOG performance status ≥ 1, the presence of ascites and residual disease after surgery, thrombocytosis, and menopausal status. These led to the development of an original prognostic score that can be helpful in clinical practice.

## 1. Introduction

Worldwide, ovarian cancer represents a significant health burden, representing the second cause of gynecological-tumor-associated mortality, according to the latest estimates of GLOBOCAN 2020 [1]. More than two-thirds of patients are diagnosed in the advanced stages of the disease because of unspecific symptoms and a lack of efficient screening and detection methods [2]. Despite adopting new treatment techniques and developing new therapeutic agents, the outcome for patients with gynecological tumors remains poor, especially in developing countries [1,3,4].

The clinician must be guided in choosing a personalized therapeutic strategy by thoroughly exploring the patients’ prognostic factors. Identifying these risk factors is vital to classify the patient in a specific risk group and estimate the risk of progression and death. These results must be correlated and interpreted in a multidisciplinary tumor board regarding selecting the correct treatment sequence between chemotherapy, targeted therapy, surgery, or radiotherapy [5,6].

A variety of adverse prognostic factors have been proposed for ovarian cancer. These include low-performance status, menopause, late-stage disease, mucinous histology and poor histologic differentiation, residual disease post-surgery, hypoalbuminemia, and thrombocytosis [7,8,9,10,11,12,13,14,15,16,17].

The most established prognostic factor in ovarian cancer is the presence of remaining cancer cells after a radical surgery aimed to remove as much of the tumor as possible. Based on the amount of residual disease, the risk can be stratified as follows: complete resection (R0): no cancer cells detectable, with best prognosis; microscopic residual disease (R1): cancer cells detectable only under the microscope, with intermediate prognosis; and macroscopic residual disease (R2): visible evidence of cancer remaining in the body post-surgery, with the worst prognosis [18]. Complete cytoreduction is associated with improved outcomes in all patients with ovarian cancer. As such, the residual disease status is a significant predictor of poor survival [10,19,20,21,22,23].

Thrombocytosis, defined as an elevated platelet count, has also been identified as a prognostic factor in patients with ovarian cancer. It is associated with advanced stages of the disease and decreased survival, as it may be caused by paracrine circuits involving thrombopoietic cytokines that stimulate tumor growth [15,16].

Another relevant marker of poor prognosis is hypoalbuminemia, defined as decreased levels of serum albumin (typically below 3.5 g/dL), and has been correlated with worse outcomes in advanced stages and may be a potential predictor of post-surgical survival [17]. A clinical surrogate for hypoalbuminemia is the presence of ascites, as low levels of albumin can cause an imbalance in oncotic and hydrostatic forces, leading to the leakage of fluids out of blood vessels into the abdominal cavity and has been associated with decreased survival and an increased risk of postoperative complications [24].

The study’s primary objectives were to assess the impact of clinical and pathological prognostic factors and establish a simple prognostic score that can help clinicians estimate patients’ risk and personalize the treatment in current medical practice. Identifying such a risk-factor-based prognostic score with predictive value in everyday routine can be helpful for clinicians and provide a meaningful benefit in treatment outcomes for ovarian cancer patients.

## 2. Materials and Methods

### 2.1. Patients

The ambispective study included 195 patients diagnosed with ovarian adenocarcinoma between 2007–2019. The study was approved by “Prof. Dr. Al. Trestioreanu” Bucharest Institute of Oncology Ethical Committee No. 22333/2022. No specific informed consent form (ICF) was used because all patients signed the institutional ICF giving consent to full use of their medical records for research purposes. The study was conducted in harmonization with the Declaration of Helsinki.

Patient’s medical records were analyzed retrospectively between 2007–2010 and prospectively between 2011–2019. Histopathological confirmation of ovarian carcinoma, stage IC-IV, good performance status (ECOG 0-2), acceptable hematologic, liver, and renal function tests to allow the treatment administration, and consent of patients were among the inclusion criteria. The exclusion criteria included ECOG ≥ 3, lack of informed consent of patients, the impossibility of delivering chemotherapy, abnormal hematologic, liver, and renal function tests, and losing contact with the patients during the follow-up period.

### 2.2. Treatment

The therapeutic strategy involved multidisciplinary decisions, including surgery followed by adjuvant chemotherapy, neoadjuvant chemotherapy followed by surgery, or only chemotherapy in selected cases (multiple comorbidities, poor general health status). Bilateral salpingo-oophorectomy, hysterectomy, and omentectomy, with or without para-aortic lymphadenectomy, were among the surgical interventions required to obtain no macroscopic residual disease, along with the possible resection of any other involved segment (bowel, diaphragm, hepatic resection, appendectomy, partial cystectomy, metastasectomy). First-line chemotherapy included at least four cycles of platinum salts doublets, and the protocols included paclitaxel 175 mg/m^2^ and carboplatin (AUC = 5) or cisplatin (75 mg/m^2^).

### 2.3. Statistical Analysis

The statistical analysis was realized with IBM SPSS, version 23.0 (Chicago, IL, USA) for Windows and Excel, and included all eligible patients. Progression-free survival (PFS) and overall survival (OS) represented the endpoints of the analysis. The Kaplan–Meier method was used for generating survival curves. The univariate analysis using the log-rank test was used for studying the influence of relevant parameters on survival and time to disease progression, and multivariate analysis was used according to the stepwise Cox proportional hazards model to identify independent prognostic factors and estimate their effect on the time to disease progression and overall survival. The confidence interval (CI) considered for the calculated quantitative variables was 95%, and the *p*-value considered statistically significant was <0.05. ROC curves were used to measure the model’s efficacy and estimate the method’s sensibility and specificity. An AUC closer to 1 is considered efficient, and AUC values > 0.6 validate the model.

## 3. Results

### 3.1. Baseline Characteristics

The median age of patients was 54 ± 10.63 years (range between 18 and 82 years).

A complete physical examination was required to assess patients’ clinical status, and the performance status was evaluated according to the ECOG scale. Most patients (61.7%) presented good performance status (ECOG 0), 35.6% with ECOG 1, and 0.7% with ECOG 2.

Most patients were postmenopausal at the time of diagnosis (63.6%). Known as a risk factor for ovarian cancer, the prevalence of nulliparity was 15.4% in our study’s cohort.

In our cohort, the stage distribution included: 9.7% in stage IC, 11.8% in location IIA, 1% in stage IIB, 9.8% in stage IIC, 4.1% in stage IIIA, 3.1% in stage IIIB, 43.6% in stage IIIC, and 16.9% in stage IV. Therefore, two-thirds of the patients (66.7%) were diagnosed with an advanced stage of the disease and metastasis. Most patients presented with large ovarian tumors, with a mean size of 87.6 ± 47.8 mm (range between 10–250 mm) (Figure 1). The CA125 level at diagnosis was elevated in 76% of patients, with a mean value of 616 ± 922 U/mL (range 4–4892 U/mL). Additionally, significant ascites (>500 mL) was observed at the diagnosis in 35.8% of patients.

The histopathological report confirmed the diagnosis of ovarian epithelial adenocarcinoma in all patients. Most tumors were included in the serous subtype (72.9%), 12.8% in the endometrioid subtype, 13.3% mucinous, and 1% were included in the clear cell carcinoma subtype. Moreover, 62.9% were poorly differentiated tumors (G3), 27.4% were moderately differentiated (G2), and 9.7% were well differentiated (G1).

A multidisciplinary team decided on the therapeutic strategy. It included surgery followed by adjuvant chemotherapy in 156 patients (80%), neoadjuvant chemotherapy followed by surgery in 30 patients (16.4%), and only chemotherapy in 3.6% of patients (poor performance status, comorbidities).

#### Oncologic Outcome and Prognostic Factors

After a median follow-up of 68 months (range 7–191), the median PFS for all stages was 32 months, and the median OS was 84 months. For stage IIIC, the median PFS was 20 months, and the median OS was 51 months, whereas for stage IV, the median PFS was 14 months, and the median OS was 40 months.

Some important risk factors were identified (Table 1 and Table 2).

ECOG performance status 1 or 2 (combined) compared to 0 was associated with poor prognostic outcome, a median PFS of 15 months vs. 60 months, and an OS of 38 vs. 112 months.

The presence of ascites represented a poor prognostic factor. PFS for patients with ascites was 22 months compared to 48 months for patients without ascites (*p* = 0.027). Moreover, OS was 44 months versus 120 months in favor of patients without ascites at the time of diagnosis (*p* = 0.005). The presence of ascites was associated with a 1.95 times higher risk of disease progression and a 2.31 times higher risk of death.

There was also a statistically significant difference regarding PFS for patients with thrombocytosis (defined as more than 450,000 platelets/mm^3^) at the time of diagnosis versus patients with a standard value of platelets. Therefore, the median PFS and OS for patients with normal values of platelets were 60 and 150 months, and for patients with thrombocytosis, median PFS and OS were only 20 and 36 months (*p* = 0.0001). The presence of thrombocytosis at the time of diagnosis was associated with a 2.48 times higher risk of disease progression and a 3.33 times higher risk of death.

The most important prognostic factor was the presence of residual disease after surgery evaluated according to the surgeon’s description and through imaging techniques (84.6% had a CT or MRI post-surgery before starting chemotherapy). In our series of patients, 49.23% of patients presented with residual disease after surgery. The presence of residual disease after surgery was associated with a shorter PFS (15 vs. 156 months, *p* = 0.0001) and statistically significant reduced OS (38 months vs. median not reached, *p* = 0.0003). Patients with residual disease after surgery had a five times higher risk of disease progression than patients without residual disease and a four times higher risk of death.

Additionally, the quantity of residual disease after surgery represents a prognostic factor. For patients without the residual disease, PFS was 156 months; for patients with residual disease less than 1 cm, PFS was 20 months; for patients with residual disease more than 1 cm, PFS was only 12 months (*p* < 0.0001). Median OS for patients without residual disease was not reached; for patients with residual disease less than 1 cm, OS was 60 months; for patients with the residual disease more than 1 cm, OS was only 34 months (*p* = 0.0002).

The most frequent histopathological subtype for postmenopausal women was high-grade serous carcinoma, whereas the incidence of the two histopathological subtypes was similar for premenopausal women. PFS was statistically significantly higher in premenopausal patients (60 vs. 24 months, *p* = 0.001), and the OS was 150 months vs. 60 months (*p* = 0.005).

### 3.2. Prognostic Score

After carefully analyzing the risk factors, the following variables were considered for establishing a prognostic score that could estimate the patient’s outcome, the necessary therapeutic strategy, and follow-up intensity:performance status ECOG ≥ 1,presence of ascites,menopausal status,residual disease after surgery,presence of thrombocytosis.

In our series of patients, 28 (14.4%) patients had no risk factors, 43 (22.1%) patients had one risk factor, 42 (21.5%) patients had two risk factors, 36 (18.5%) had three risk factors, 40 (20.6%) had four risk factors, and 6 (3%) had five risk factors at diagnosis.

Each additional risk factor contributed to a statistically significant reduction in PFS (*p* = 0.0001) and OS (*p* = 0.001). Patients with no risk factors and a score of 0 had not reached median PFS and OS (Figure 2a for PFS and Figure 2b for OS). Patients with one risk factor had a median PFS of 70 months and an estimated median OS of 108 months. Patients with two risk factors had a median PFS of 36 months and a median OS of 77 months. Patients with three risk factors had a median PFS of 20 months and a median OS of 60 months. Patients with four or more poor prognostic factors had a median PFS of 12 months and an estimated OS of 34 months. These data sustain the use of the clinical prognostic score in the patient’s initial evaluation.

ROC curves were used to validate the prognostic score and to characterize its sensibility and specificity. The prognostic score has an area under the curve of 0.799 (*p* = 0.0001, 95%CI 0.721–0.86) for PFS and an area under the curve of 0.726 (*p* = 0.0001, 95%CI 0.710–0.850) for OS. Therefore, these results demonstrate that this prognostic score could be helpful in medical practice due to its good sensibility and specificity (Figure 3a for PFS and Figure 3b for OS).

## 4. Discussion

Despite using new therapeutic agents and developing new treatment techniques, the outcome for patients with gynecological tumors remains poor, especially in developing countries [1,3]. Correctly identifying prognostic factors in ovarian cancer is vital, supporting the clinician in classifying the patient in a specific risk group and estimating the risk of disease progression and death. Furthermore, correctly identifying all prognostic factors leads to the right choice of therapeutic strategies and can help choose a personalized treatment. This can be achieved by associating standard treatment with additional therapeutic options such as antiangiogenic agents, PARP inhibitors, targeted therapy, immunotherapy, or radiotherapy. Identifying risk factor sets with predictive value in everyday practice can be a helpful tool.

The 195 patients with ovarian carcinoma included in this study had an extended median follow-up (68 months, range 7–191 months), allowing the identification of several adverse prognostic factors. Thus, the main prognostic factors were the stage of the disease, ECOG performance status 1 or 2 compared to 0, the presence of ascites, the degree of tumor differentiation, serous histopathological subtype compared to non-serous tumors, and menopausal status. The most important prognostic factor in our study was the presence and the amount of residual disease after surgery.

To our knowledge, an integrated prognostic score correlating these factors is yet to be developed. As such, we consider that this comprehensive method of calculating a predictive clinical marker is a novelty that the clinician needs.

The most robust data regarding the identification of prognostic factors in ovarian carcinoma originate from the analysis of three prospective studies (AGO-OVAR 3, 5, 7) conducted in Europe. The studies’ initial purpose was to evaluate the role of a third chemotherapeutic agent (epirubicin or topotecan) added to the classic combination of paclitaxel and carboplatin/cisplatin [7,8,9].

Considering the negative result of the study, researchers proposed a prospective analysis of several prognostic factors for patients with ovarian carcinoma. The study included 3126 patients and proved through a univariate and a multivariate analysis that age, FIGO stage IIIC and IV compared to stage IIIB tumors or earlier stages, grade G2 and G3 compared to G1, mucinous histopathological subtype versus serous, ECOG 2 performance status versus ECOG 0, and the presence of ascites over 500 mL are independent prognostic factors.

The study showed that only a third of the included patients met the criteria for complete resection. The minimum residual disease was defined as a tumor between 1 and 10 mm, and macroscopic residual disease was defined as a tumor over 1 cm. The complete resection of the tumor was associated with a reduction in disease progression and death of 66% and 68%, respectively. The benefit of radical surgery was maintained for all stages of the disease.

Similar results were reported by a study including 1895 patients with stage III disease [10]. Following R0 resection as a comparison standard, the study observed that the presence of residual disease between 0–10 mm or over 10 mm was associated with a rise in the risk of disease progression HR = 1.96 (95%CI, 1.70–2.26; *p* = 0.001) and HR = 2.36 (95%CI 2.04–2.73; *p* = 0.001). Moreover, the risk of death was also increased by the presence of residual disease less than 1 cm (HR = 2.11; *p* = 0.001) or over 1 cm (HR = 2.47; *p* = 0.001) compared to those without residual disease or those with macroscopic residual disease. The study concluded that age, performance status, histopathology (mucinous vs. serous and endometrioid), and residual disease after surgery were poor prognostic factors.

Another study was published in Gynecologic Oncology Journal, and the objective was to evaluate the impact of cytoreduction and residual disease in 326 patients with stage IV ovarian adenocarcinoma. Optimal surgical resection was obtained in 54.9% of patients, 30.8% presented residual disease between 1–10 mm, and 14.3% showed residual disease over 10 mm after cytoreduction. The median OS was statistically significantly better for patients who underwent optimal cytoreduction without the residual disease (50 months) compared to patients with the residual disease between 1–10 mm when OS was 25 months or patients with the residual disease over 1 cm when OS was only 16 months.

The multivariate analysis confirmed the inferiority regarding the lack of surgical intervention (HR = 2.51, *p* = 0.0001), of residual disease between 1 and 10 mm (HR =1.5, *p* = 0.046), and residual disease over 1 cm (HR = 2.17, *p* = 0.002) compared to optimal cytoreduction, without the macroscopic residual disease. The study identified other prognostic factors, such as performance status, presence of ascites over 500 mL, and extension of the disease to the abdominal wall or liver metastasis, which are associated with a reserved prognosis [19].

Even more recent studies have elaborated a predictive model of response after neoadjuvant chemotherapy and discovered that patients with complete or near complete response (CRS3) (28%) after platinum salts doublet had a better PFS and OS compared to patients with partial CRS2 or no/minimal response CRS1 (HR = 0. 55, 95%CI, *p* = 0.001 for PFS and 0.65, *p* = 0.002 for OS) [20]. A similar study found that these data are valid for residual ovarian tumors and omental metastasis [21]. A recent systematic review identified that the Peritoneal Cancer Index (PCI) with an AUC of 0.69–0.92 [22] and Predictive Index Value (PIV) with an AUC of 0.66–0.98 were the most critical scores for complete resection [23].

Menopausal status is a known factor associated with ovarian adenocarcinoma, with over 65% of patients being postmenopausal at diagnosis [11]. Limited data are mentioned in the literature regarding ovarian adenocarcinoma in premenopausal patients; thus, the diagnosis is delayed in most cases. The proportion of young patients in our study is similar to the data mentioned in the literature [12,13].

A study that included 496 patients with malignant epithelial tumors reported an increased incidence of ovarian carcinoma with aging. Similar to our results, the percentage of patients with endometrioid carcinoma was higher in young patients compared to postmenopausal patients [14]. Although most patients are diagnosed during menopause, the reproductive characteristics of the patients, such as nulliparity or reduced number of pregnancies and the use of contraceptive pills, represent the most important risk factors [25]. Nulliparous patients had a higher risk of ovarian cancer than patients who gave birth (HR = 0.69, *p* = 0.001) and a significantly higher risk of clear cell carcinoma (RR = 0.35). In contrast, patients with serous carcinoma had the lowest risk reduction (RR = 0.81) [26].

Another important prognostic factor identified in our patients was the presence of thrombocytosis at diagnosis. In a study including 619 ovarian carcinoma patients, the aim was to establish the relationship between the number of thrombocytes in the peripheral blood smear, time to disease progression, and overall survival of patients. The study demonstrated that thrombocytosis is associated with advanced stages of the disease and decreased survival. Moreover, a rise in thrombopoietin values and IL-6 was observed, which explains the existence of a paracrine circuit where the increase in thrombopoietic cytokines leads to paraneoplastic thrombocytosis, which stimulates tumor growth [15].

A study that included more than 100 patients with ovarian cancer considered the value of pretreatment platelet count and tumor markers such as CA125 and proposed a reliable and straightforward to use in clinical practice score for patients with stage IV disease. A combined platelet and CA125 score of 0, 1, and 2 were determined based on the presence of thrombocytosis defined as more than 400,000/μL, elevated CA125 level defined as more than 1200 U/mL, or both. Median PFS was significantly lower in patients with a Platelet-CA125 score of 2 (19.6 months) compared with patients with a score of 0 (32.0 months; *p* = 0.011). Multivariate analysis identified both Platelet-CA125 scores of 2 and 1 as independent poor prognostic factors both for overall survival (*p* = 0.004, *p* < 0.001) and progression (*p* = 0.033, *p* = 0.017) in comparison with a score of 0 [16].

Another study aimed to validate a nomogram that predicts the 3-year recurrence risk of ovarian carcinoma. The items included in the nomogram were FIGO stage, histological grade, histological type, lymph node metastasis status, and serum CA125 level at diagnosis. The ROC curve of the nomogram showed that the AUC was 0.828 and identified a threshold value with reasonable specificity and sensitivity [27]. The study also showed that patients with thrombocytosis (53%) had a statistically significant lower PFS than those with a normal platelet count. OS was 2.62 years for patients with thrombocytosis and 4.65 years for patients with a normal thrombocyte count. The multivariate analysis showed that by analyzing according to age, stage of disease, grading, histopathological subtype, and residual disease after surgery, thrombocytosis remains an adverse prognostic factor for survival (*p* < 0.001).

Hypoalbuminemia at diagnosis was identified as a poor prognostic factor in advanced stages. More than that, after neoadjuvant treatment, if albumin levels return to normal, this may be a potential predictor of survival after surgery [17].

Markers of angiogenesis such as vascular endothelial growth factors VEGF and oxidative stress markers such as malondialdehyde may be used as prognostic markers, but their use on a large scale is limited [6,28,29,30,31].

In recurrent ovarian cancer, too, such a risk score that included patient’s characteristics (ECOG performance status, age, quality of life, and nausea/vomiting) and treatment characteristics, such as platinum-free interval, showed a good predictive value with an AUC of 0.81 [32].

The future will belong to risk scores that include gene signatures. Such a score will be developed and validated based on the expression and augmentation of ovarian-cancer-related genes used to predict the outcome and chemoresistance in ovarian cancer patients [33].

New data are expected from the OTTA-SPOT (Ovarian Tumor Tissue Analysis consortium—Stratified Prognosis of Ovarian Tumors). They developed 276 gene expression signatures, identifying patients likely to achieve 5-year survival [34].

## 5. Conclusions

Several negative prognostic factors were identified: ECOG performance status ≥ 1, ascites, the presence and quantity of residual disease after surgery, thrombocytosis, and menopausal status. These results led to the development of an original prognostic score, which can be useful in clinical practice. Therefore, the clinical prognostic score could allow medical oncologists and surgeons to identify patients with adverse prognostic factors, for which treatment should be individualized through the escalation of therapeutic strategies.

## Figures and Tables

**Figure 1 medicina-59-00229-f001:**
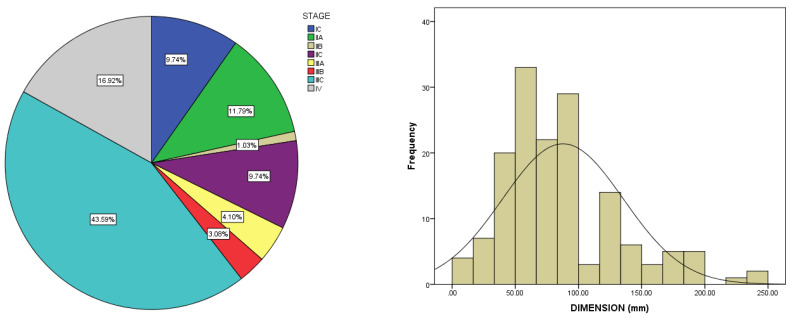
Tumor characteristics according to International Federation of Gynecology and Obstetrics (FIGO) staging and tumor dimension.

**Figure 2 medicina-59-00229-f002:**
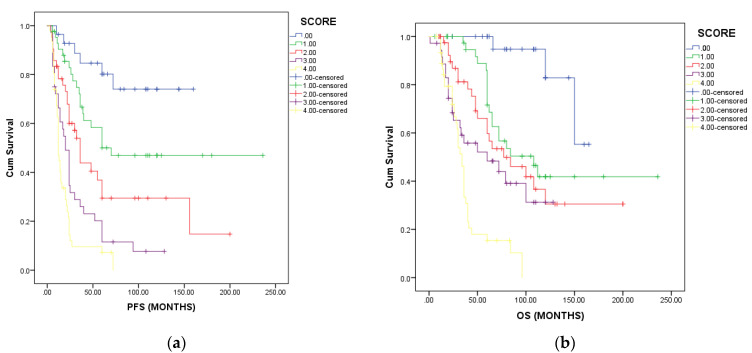
Progression-free survival (**a**) and overall survival (**b**) according to the number of risk factors.

**Figure 3 medicina-59-00229-f003:**
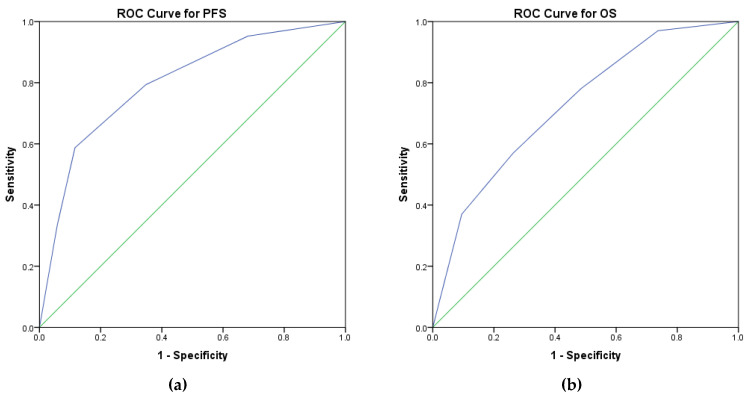
ROC curves estimating the specificity and sensitivity of the prognostic score ((**a**)-PFS and (**b**)-OS).

**Table 1 medicina-59-00229-t001:** Progression-free survival (PFS) and overall survival (OS) according to prognostic factors.

	PFS		OS	
Characteristics	Median PFS (Months)	*p*	Median OS (Months)	*p*
Eastern Cooperative Oncology Group (ECOG)				
ECOG 0	60		112	
ECOG 1	20	0.002	40	0.0001
ECOG 2	10	0.001	30	0.0001
Menopause				
No	60		150	
Yes	24	0.001	60	0.005
Residual disease				
Yes	15		38	
No	156	<0.0001	NR	0.0003
R1 cm				
0	156		NR	
0–1 cm	20	<0.0001	60	0.0002
More 1 cm	12	<0.0001	34	0.0001
G				
G2	72		150	
G3	21	<0.0001	50	0.0001
Histopathology (HP)				
Serous	24	0.002	62	NS
Mucinous	60	NS	72	
Endometrioid	70	NS	110	
Clear Cell	30	NS	NR	
Ascites				
Yes	22		44	
No	48	0.027	120	0.005
Thrombocytosis				
Yes	60		36	
No	20	0.0001	152	0.0001

**Table 2 medicina-59-00229-t002:** Multivariate analysis of the prognostic factors for PFS and OS.

	PFS			OS		
Characteristics	HR	PFS 95%CI	*p*	HR	OS 95%CI	*p*
ECOG						
ECOG 0	1			1		
ECOG 1, 2	2.71	1.96–3.73	0.001	3.19	2.20–4.64	0.001
Thrombocytosis						
No	1			1		
Yes	2.48	1.72–3.58	0.0001	3.33	2.16–5.13	0.0001
Menopause						
No	1			1		
Yes	2.02	1.35–3.01	0.001	2.25	1.41–3.59	0001
R0						
Yes	1			1		
No	5.12	3.43–7.65	0.0001	4.07	2.59–6.39	0.0001
G						
G2	1			1		
G3	2.50	1.77–3.53	0.001	2.24	1.65–3.24	0.001
HP						
Serous	1			1		
Non-serous	0.68	0.61–0.90	0.008	0.56	0.5–1	NS
Ascites						
No	1			1		
Yes	1.95	1.35–2.80	0.03	2.31	1.52–3.5	0.007

## Data Availability

All data generated and analyzed are included within this research article.

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
