# Peer review of "Validation of a New Prognostic Score in Patients with Ovarian Adenocarcinoma"

_medicina, 2023, doi:10.3390/medicina59020229_

Round 1
Reviewer 1 Report
Dear Authors,
I am enthusiastic to review your manuscript, focused on a topic of interest to me, and providing research data that is in line with the current developments in the field. I would like to take this chance to also congratulate you for your work and the effort put into the validation of a new prognostic score in patients with epithelialovarian adenocarcinoma.
In the review process, I would like to point out aspects that might bring a slight improvement to your manuscript in the following section-by-section assessment:
The Simple Summary and Abstract serve their purpose and are generally well written. However, lines 53-54 could use rephrasing, due to their ambiguousness.
Introduction:
While clear and concise, it could briefly describe the pathology and how the negative prognosis is currently assessed. Currently, it is a motivation statement rather than a background section with up-to-date information introducing the reader to the topic. In the manuscript, the team has covered this in the Discussion section. However, the introduction could contain more references that are relevant and new.
Results:
The median age is not as indicative as a mean ± SD in showing the age distribution. Please consider providing.
Lines 127-134 could be complemented with graphs. This would increase readability.
Line 146, 147: mean ± SD to be added
Line 210: diagnosis, not diagnostic
Note: the median does not clearly reflect the interval/distribution of the results. Please add the mean and standard deviation in all instances. In my opinion, this needs to be extended to the calculations.
Discussion:
The discussion could be restructured. Presenting the data of previous studies in review style without comparing and contrasting or complementing original results of the authors’ work is not fitting the context. While the information is there, please include it in the context of your research, as this is not a review.
With these changes, I do consider the work valuable and fitting the standards for publication.
Best regards,
Your Reviewer
As a note, the references 6, 18 and 29 do not seem as the most appropriate works in the field to be cited, however, as the authors are most familiar with the work they have contributed to, it could be acceptable in my view.
Author Response
Dear Editor,
We thank you for allowing us to submit a revised draft of our manuscript titled “Validation of a new prognostic score in patients with ovarian adenocarcinoma” to the Medicina Journal. We thank you and the reviewers for the insightful comments and kind suggestions, which helped us significantly improve our article. Starting from the missing details they observed, we made substantial changes and believe the issues being discussed have been substantially clarified. Please find the modifications marked in red in the revised manuscript.
Here is a point-by-point response to the reviewers’ comments and concerns.
Response to Reviewer 1 Comments:
We are deeply for your insightful feedback and kind words of encouragement! Your points helped us develop a better version of our manuscript. Here is a point-by-point description of the changes we have made.
Point 1: lines 53-54 could use rephrasing, due to their ambiguousness
Response for Point 1: Lines 53-54 were rephrased to “The median PFS for patients with one, two, three, or four (or more) prognostic factors was not reached at 70, 36, 20, and 12 months, respectively. The corresponding median overall survival (OS) was not reached at 108, 77, 60, and 34 months, respectively.”
Point 2: Introduction: While clear and concise, it could briefly describe the pathology and how the negative prognosis is currently assessed. Currently, it is a motivation statement rather than a background section with up-to-date information introducing the reader to the topic. In the manuscript, the team has covered this in the Discussion section. However, the introduction could contain more references that are relevant and new.
Response for Point 2:
We want to thank you for this feedback! We decided to overhaul the introduction completely. Written below is our new version:
Worldwide, ovarian cancer represents a significant health burden, representing the second cause of gynecological tumors-associated mortality, according to the latest estimates of GLOBOCAN 2020 [1]. More than two-thirds of patients are diagnosed in the advanced stages of the disease because of unspecific symptoms and a lack of efficient screening and detection methods [2]. Despite adopting new treatment techniques and developing new therapeutic agents, the outcome of patients with gynecological tumors remains poor, especially in developing countries [1,3,4].
The clinician must be guided in choosing a personalized therapeutic strategy by thoroughly exploring the patients’ prognostic factors. Identifying these risk factors is vital to classify the patient in a specific risk group and estimate the risk of progression and death. These results must be correlated and interpreted in a multidisciplinary tumor board regarding selecting the correct treatment sequence between chemotherapy, targeted therapy, surgery, or radiotherapy [5,6].
A variety of prognostic adverse prognostic factors have been proposed for ovarian cancer. These include low-performance status, menopause, late-stage disease, mucinous histology and poor histologic differentiation, residual disease post-surgery, hypoalbuminemia, and thrombocytosis [7–17].
The most established prognostic factor in ovarian cancer is the presence of remaining cancer cells after a radical surgery to remove as much of the tumor as possible. Based on the amount of residual disease, the risk can be stratified as follows – complete resection (R0): no cancer cells detectable, with best prognosis; microscopic residual disease (R1): cancer cells detectable only under the microscope, with intermediate prognosis; and macroscopic residual disease (R2): visible evidence of cancer remaining in the body post-surgery, with the worst prognosis [18]. Complete cytoreduction is associated with improved outcomes in all patients with ovarian cancer. As such, the residual disease status is a significant predictor of poor survival [10,19–23].
Thrombocytosis, defined as an elevated platelet count, has also been identified as a prognostic factor in patients with ovarian cancer. It is associated with advanced stages of the disease and decreased survival, as it may be caused by paracrine circuits involving thrombopoietic cytokines that stimulate tumor growth [15,16].
Another relevant marker of poor prognosis is hypoalbuminemia, defined as decreased levels of serum albumin (typically below 3.5 g/dL), and has been correlated with worse outcomes in advanced stages and may be a potential predictor of post-surgical survival [17]. A clinical surrogate for hypoalbuminemia is the presence of ascites, as low levels of albumin can cause an imbalance in oncotic and hydrostatic forces, leading to leakage of fluids out of blood vessels into the abdominal cavity and has been associated with decreased survival and increased risk of postoperative complications [24].
The study's primary objectives were assessing the impact of clinical and pathological prognostic factors and establishing a simple prognostic score that can help clinicians estimate patients’ risk and personalize the treatment in current medical practice. Identifying such a risk factor-based prognostic score with predictive value in everyday routine can be helpful for clinicians and provide a meaningful benefit in treatment outcomes for ovarian cancer patients.
Point 3: The median age is not as indicative as a mean ± SD in showing the age distribution. Please consider providing. Note: the median does not clearly reflect the interval/distribution of the results. Please add the mean and standard deviation in all instances. In my opinion, this needs to be extended to the calculations.
Response to point 3: Standard and mean deviation added in all instances.
Point 4: Lines 127-134 could be complemented with graphs. This would increase readability.
Response to Point 4: We added Figure 1.
Figure 1. Tumor characteristics according to FIGO staging and tumor dimension
Point 5: Line 146, 147: mean ± SD to be added
Response to Point 5: Done. Thank you for pointing out.
Point 6: Line 210: diagnosis, not diagnostic
Response to Point 6: We made the change. We also made corrections to our grammar, spelling and editing, per your suggestion.
Point 7: The discussion could be restructured. Presenting the data of previous studies in review style without comparing and contrasting or complementing original results of the authors’ work is not fitting the context. While the information is there, please include it in the context of your research, as this is not a review.
Response to Point 7: We compare the results obtained with the one in the literature
We added the following paragraph:
Lines 491 – 493: To our knowledge, an integrated prognostic score correlating these factors is yet to be developed. As such, we consider that this comprehensive method of calculating a predictive clinical marker is a novelty that the clinician needs.
We thank the Reviewer for very clear and detailed feedback on our manuscript, which allowed us to write this improved version!

Reviewer 2 Report
The authors present an interesting article suggesting a new prognostic score in patients with ovarian adenocarcinoma.
There are some issues that should be explained
Major issues:
1) In lines 135-140, the authors describe the histopathology of the tumors as “The histopathological report confirmed the diagnosis of epithelial ovarian adenocarcinoma in all patients. Most tumours were included in the papillary serous subtype (72.9%), 12.8% in the endometrioid subtype, 13.3% mucinous and 1% were included in the clear cell carcinoma subtype. Moreover, 52.1% were poorly differentiated tumours (G3), 38.2% were moderately differentiated (G2) and 9.7% were well differentiated (G1)”.
Since the majority of tumors are serous carcinomas (the term papillary serous is no longer used), accounting for almost 73% of the total number of cases, it is surprising that only 52.1% of tumors are poorly differentiated. At least some cases of serous carcinoma must have been assigned as grade 2. Serous carcinomas are either high or, less frequently, low-grade tumors. How were these tumors graded? I believe that if the authors grade serous carcinomas as either high or low grade, maybe the results of their study considering the importance of grading could be different.
2) In lines 179-180, it is stated that “the presence of residual disease after surgery evaluated according to the surgeon’s description or through imaging techniques”. In how many cases was the surgeon’s description used as the only method to evaluate the presence of residual disease? Does this mean that in these cases, there were no imaging techniques performed? Please clarify.
Minor issues:
1) The authors should consider removing the word epithelial from the title since all adenocarcinomas are epithelial tumors.
Author Response
Dear Editor,
We thank you for allowing us to submit a revised draft of our manuscript titled “Validation of a new prognostic score in patients with ovarian adenocarcinoma” to the Medicina Journal. We thank you and the reviewers for the insightful comments and kind suggestions, which helped us significantly improve our article. Starting from the missing details they observed, we made substantial changes and believe the issues being discussed have been substantially clarified. Please find the modifications marked in red in the revised manuscript.
Here is a point-by-point response to the reviewers’ comments and concerns.
We want to thank you for your inspiring and insightful remarks that determined us to improve our manuscript. As per your recommendation to make English language modifications, we have made several changes that have been tracked. Here is our point-by-point description of the changes we have made.
Point 1: In lines 135-140, the authors describe the histopathology of the tumors as “The histopathological report confirmed the diagnosis of epithelial ovarian adenocarcinoma in all patients. Most tumours were included in the papillary serous subtype (72.9%), 12.8% in the endometrioid subtype, 13.3% mucinous and 1% were included in the clear cell carcinoma subtype. Moreover, 52.1% were poorly differentiated tumours (G3), 38.2% were moderately differentiated (G2) and 9.7% were well differentiated (G1)”. Since the majority of tumors are serous carcinomas (the term papillary serous is no longer used), accounting for almost 73% of the total number of cases, it is surprising that only 52.1% of tumors are poorly differentiated. At least some cases of serous carcinoma must have been assigned as grade 2. Serous carcinomas are either high or, less frequently, low-grade tumors. How were these tumors graded? I believe that if the authors grade serous carcinomas as either high or low grade, maybe the results of their study considering the importance of grading could be different.
Response for Point 1: We deleted “papillary serous,” and we have modified lines 135-140:
The histopathological report confirmed the diagnosis of ovarian epithelial adenocarcinoma in all patients. Most tumors were included in the serous subtype (72.9%), 12.8% in the endometrioid subtype, 13.3% mucinous, and 1% were included in the clear cell carcinoma subtype. Moreover, 62.9% were poorly differentiated tumors (G3), 27.4% were moderately differentiated (G2), and 9.7% were well differentiated (G1).
Our pathologist graded some of the samples as G2-G3. We decided to consider these patients as G3.
Point 2: In lines 179-180, it is stated that “the presence of residual disease after surgery evaluated according to the surgeon’s description or through imaging techniques.” In how many cases was the surgeon’s description used as the only method to evaluate the presence of residual disease? Does this mean that in these cases, there were no imaging techniques performed? Please clarify.
Response for Point 2: We decided to rephrase lines 179-189 to:
The most important prognostic factor was the presence of residual disease after surgery evaluated according to the surgeon’s description and through imaging techniques (84.6% had a CT or MRI post-surgery before starting chemotherapy).
Point 3: The authors should consider removing the word epithelial from the title since all adenocarcinomas are epithelial tumors.
Response for Point 3: Title modified, removed the word epithelial.
New title: Validation of a new prognostic score in patients with ovarian adenocarcinoma
Again, we thank you for your comments and suggestions,
The authors

Round 2
Reviewer 1 Report
Dear Authors,
Thank you for implementing the changes suggested by all reviewers and responding to the letters.
I find the article to have been significantly improved since the last correction.
Thank you for your scientific contribution!
Reviewer 2 Report
The authors have made all the changes I have asked for. The manuscript is O.K., in my opinion.